

# Improving fraud detection with semi-supervised topic modeling and keyword integration

Marco Sánchez[1] and Luis Urquiza[2]

[1] Departamento de Informática y Ciencias de la Computación, Escuela Politécnica Nacional, Quito, Pichincha, Ecuador

[2] Departamento de Electrónica, Telecomunicaciones y Redes de Información, Escuela Politécnica Nacional, Quito, Pichincha, Ecuador

## ABSTRACT

Fraud detection through auditors' manual review of accounting and financial records has traditionally relied on human experience and intuition. However, replicating this task using technological tools has represented a challenge for information security researchers. Natural language processing techniques, such as topic modeling, have been explored to extract information and categorize large sets of documents. Topic modeling, such as latent Dirichlet allocation (LDA) or non-negative matrix factorization (NMF), has recently gained popularity for discovering thematic structures in text collections. However, unsupervised topic modeling may not always produce the best results for specific tasks, such as fraud detection. Therefore, in the present work, we propose to use semi-supervised topic modeling, which allows the incorporation of specific knowledge of the study domain through the use of keywords to learn latent topics related to fraud. By leveraging relevant keywords, our proposed approach aims to identify patterns related to the vertices of the fraud triangle theory, providing more consistent and interpretable results for fraud detection. The model's performance was evaluated by training with several datasets and testing it with another one that did not intervene in its training. The results showed efficient performance averages with a 7% increase in performance compared to a previous job. Overall, the study emphasizes the importance of deepening the analysis of fraud behaviors and proposing strategies to identify them proactively.

# INTRODUCTION

Auditors can identify fraud by reviewing accounting and financial records; their experience can detect this phenomenon. Reproducing this task using technological tools has become a challenge for researchers in computer security. There are several initiatives to transfer auditors' knowledge to the technical area by applying Machine learning techniques and theories related to fraud. However, representing intuitive expert judgments means a challenge, especially when the result of applying formal methodologies does not coincide with the experts' criteria. By applying topic modeling, it is possible to codify human

Corresponding author
Marco Sánchez,
marco.sanchez01@epn.edu.ec

knowledge and then use it to extract interpretable latent topics from a corpus. Topic modeling was introduced in an unsupervised environment (*Blei, Ng & Jordan, 2003*), the most conventional being Dirichlet assignment (LDA) or non-negative matrix factorization (NMF), which have become popular in recent years. These are based on statistical models that allow discovering thematic structures in collections of texts, identifying themes humans can interpret and facilitating their understanding (*Egger & Yu, 2022*). The structures found by unsupervised topic modeling often do not represent the best alternative for analyzing a specific phenomenon.

A topic modeler trained in reviewing documents in a corpus can be considered when this can discover the implicit semantic structures that describe general topics. However, we often want to dig deeper by discovering topics that reflect a specific behavior, in our case, related to fraud. Techniques for effectively finding patterns related to topics linked to a particular field are called semi-supervised topic modeling, which generates interpretable topics. For this reason, we propose using this modeling technique to learn latent issues about documents. Unlike LDA, these do not assume a specific data generation model and instead, look for "more informative" topics.

In this context, we analyze in this article the most used semi-supervised topic models, such as correlation explanation (CorEx) (*Gao et al., 2019*) and Seeded LDA, to validate their performance in a detector of suspected fraud behavior (*Sánchez-Aguayo, Urquiza-Aguiar & Estrada-Jiménez, 2022*) that analyzes human behavior using the fraud triangle theory (FTT) (*Fitri, Syukur & Justisa, 2019*) leveraged in machine learning (ML).

This will allow for flexibly incorporating the knowledge of the domain of study through the use of keywords within the topic modeling, which can lead the experimentation toward discovering topics that otherwise would remain hidden.

Using relevant words can help our proposed detector recognize patterns related to the vertices of the fraud triangle, thus allowing the analysis to be guided directly on this fraud theory.

## Related work

Semi-supervised topic modeling has been the focus of various research studies in domains such as clinical notes and marketing. These studies have proven to be valuable as they offer topics that are easily interpretable. For example, *Pecòre (2021)* utilized the Anchored Correlation Explanation (CorEx) algorithm to extract English tweets related to eating disorders, aiming to develop a tool for identifying this disorder. Another study by *Shamna, Govindan & Nazeer (2019)* introduced an automated medical image retrieval system incorporating subject and location probabilities to enhance performance. Using the guided latent Dirichlet assignment (GuidedLDA), method facilitated the generation of topic information. This approach demonstrated superior average mean precision (86.74) and precision (97.5) compared to previous methods.

*Lyall-Wilson, Kim & Hohman (2019)* suggested using topic modeling to identify human factors-related topics in aviation safety reports. Utilizing algorithms like CorEx and SeededLDA achieved more accurate results without requiring manual revisions. Similarly, *Reing et al. (2016)* also explored using the CorEx algorithm for topic modeling,

aiming to extract interpretable latent topics by harnessing informal human knowledge. The study by *Koh & Fienup (2021)* employed various topic modeling techniques to analyze chat data collected in a library to extract specific and easily interpretable topics. They evaluated the results quantitatively using the coherence metric, while a librarian who was also an author of the article assessed qualitative accuracy and interpretability. *Gallagher et al. (2017)* presented a topic-modeling approach incorporating relevant words to identify rare diseases not mentioned in clinical health notes. The objective was to provide relief workers with better guidance in offering practical help and eliminating ambiguities when analyzing complex problems.

In a recent study, *Steuber, Schoenfeld & Rodosek (2020)* utilized topic modeling to evaluate semantic relationships in short messages on Twitter. They could identify associations with specific discussion topics by analyzing the hashtags used in these messages. This method proved helpful in understanding the content and context of these messages. *Olivier et al. (2019)* took a qualitative approach and developed a natural language processing tool called guided latent Dirichlet allocation (GLDA). This tool analyzed entertainment products, such as award-winning films, based on media psychology literature. By predicting viewers' behavior, they demonstrated the potential of this approach for understanding consumer behavior in film selection. *Hoffmann, Shi & Rüppel (2021)* focused on generating new document classification systems using automatic learning methods. They employed LDA to identify groups of words related to the attributes of the documents, enabling efficient document search based on matching keywords by topic.

To address the issue of overlapping topics, they utilized guided LDA, which allowed them to influence topic generation by setting seed words per topic. Another study by *Ferner et al. (2020)* proposed a method to automatically identify seed words for disaster-related topics. By comparing words from tweets on the day of the disaster occurrence with the previous day in the same area, they could obtain initial words using LDA. These words were then used to identify tweets related to the event. This method proved effective in automatically identifying relevant words for disaster-related topics. *Egger & Yu (2021)* evaluated different topic modeling algorithms for knowledge extraction in the tourism industry. Their findings showed the complexity of analyzing short-text social media data and emphasized the effectiveness of using CorEx to analyze Instagram content. CorEx outperformed LDA and NMF in ranking relevant sites and activities. LDA results were homogeneous and overlapping, while topics extracted from NMF were not specific enough to gain deep insights.

These research works demonstrate the diverse applications and benefits of semi-supervised topic modeling in different domains. Using algorithms like CorEx and GuidedLDA allows for more precise and interpretable topic extraction; This enhances our understanding of complex topics and enables the development of practical tools for identifying specific disorders, improving medical image retrieval systems, and analyzing human factors in safety reports.

Additionally, Table 1 presents a summary where it is provided information includes methods used, publication year, fields, and purpose to the significant state of the art.

**Table 1 Research papers grouped by topics and field.**

| Topic | Field | Authors | Method used | Purpose/Outcome |
|---|---|---|---|---|
| Medical and healthcare | Clinical notes | – | Semi-supervised topic modeling | Extract interpretable topics |
| | Eating disorders | *Pecóre (2021)* | Anchored CorEx | Identify eating disorders |
| | Medical image retrieval | *Shamna, Govindan & Nazeer (2019)* | GuidedLDA | Improve image retrieval |
| | Rare disease recognition | *Gallagher et al. (2017a)* | Topic modeling | Recognize rare diseases |
| Human factors and safety | Aviation safety | *Lyall-Wilson, Kim & Hohman (2019)* | CorEx, SeededLDA | Identify human factor-related topics |
| Social media analysis | Human knowledge | *Reing et al. (2016)* | CorEx | Extract informal human knowledge |
| | Semantic relationships | *Steuber, Schoenfeld & Rodosek (2020)* | Topic modeling | Analyze semantic relationships |
| Cultural and entertainment | Entertainment description | *Olivier et al. (2019)* | Guided LDA | Analyze films and predict behavior |
| Information retrieval | Document classification | *Hoffmann, Shi & Rüppel (2021)* | Topic models with metadata | Enable document search using topics |
| Disaster and event related | Disaster identification | *Ferner et al. (2020)* | LDA | Identify disaster-related topics |
| Tourism and social media | Tourism knowledge | *Egger & Yu (2021)* | Topic modeling | Analyze tourism content |
| Library and information | Library chats | *Koh & Fienup (2021)* | Various topic modeling | Analyze library chat data |

A detailed study on fraud-related jobs was conducted in *Sánchez-Aguayo, Urquiza-Aguiar & Estrada-Jiménez (2021)*. A systematic literature review (SLR) proposes collecting and analyzing research that addresses this phenomenon, considering human behavior as the leading risk factor reviewed associated theories that study this phenomenon. In addition, Machine learning techniques were incorporated into the research that allows their detection.

This work was developed in the context of a previous investigation entitled "Predictive Fraud Analysis Applying the FTT through Data Mining Techniques" (*Sánchez-Aguayo, Urquiza-Aguiar & Estrada-Jiménez, 2022*). They propose a detector of suspected fraud behavior by analyzing human behavior using the FTT leveraged in machine learning (ML) and deep learning (DL). To develop this proposal, they evaluated the performance of frequently used text mining techniques, such as LDA, NMF, and latent semantic analysis (LSA). Finally, to determine the differences in performance, they used receiver operating characteristic (ROC) curves based on the area under the curve (AUC) with the traditional ML classification methods to identify which technique is more compatible to work with the modeling of topics to detect suspicious behavior of fraud.

In this context, the present work proposes to deepen the analysis of topic modeling through the use of semi-supervised techniques associated with fraud theories that, through classification algorithms, make it possible to more efficiently detect possible cases of fraud

not observed in the works mentioned above. Therefore, this represents a clear research gap in this area.

### Contribution

The main contributions are the following: first, we use CorEx as a topic model and perform an efficient alteration of its code to identify the probabilities that the corpus documents belong to a topic and to be able to visualize the distribution of topics through the pyLDAvis library. Second, we show how the FTT can be integrated into CorEx through ''keywords'' related to the vertices of this theory. We show that CorEx produces more relevant topics than its unsupervised and semi-supervised variants of LDA.

Once the most efficient semi-supervised topic modeling has been identified, the probabilities that a document belongs to a specific topic are obtained, with which classification methods such as gradient boosting (GB) and random forest (RF) were trained to try to predict related cases with fraud. Finally, the proposed model is validated with the different datasets used in this research to try to establish the generality of the model.

Several synthetic datasets were used and generated to validate their performance to ensure the model's accuracy. The datasets were generated using various techniques to simulate different scenarios and environments. The model was tested in multiple conditions to ensure it worked reliably in all situations, confirming that it could accurately predict outcomes in various contexts. The results of these tests were then used to validate the model's performance and provide evidence of its accuracy.

The rest of this document is organized as follows: The ''Background'' section provides relevant information on FTT, topic modeling, and machine learning classification methods. Then, the Section ''Methodology'' describes the data preparation and the methodology used in this work. Next, the Section ''Results and Discussion'' deals with the experiment, the results, the validation, and the discussion. Finally, the ''Conclusions'' section is presented, addressing future work.

## BACKGROUND

This section briefly describes the FTT, topic modeling strategy, classification, and validation methods.

### Fraud theories

Today's society is constantly changing due to factors like globalization, technological advancements, and the rapid growth of industries. This creates several difficulties, particularly those about information security and management. Due to this, there may be an increase in fraud risk for both public and private companies. Organizations are now more aware of the need for fraud detection and prevention techniques due to the high crime rates to reduce the risk of fraud (*Fitriyah & Novita, 2021*). Organizations face a severe problem with cybersecurity and the risks that come with it due to internal and external factors worldwide. The internal ones are related to the companies' inherent management and commercial activity, while the external ones are related to politics and

the global economy. These risks exist, increasing the chance that they could become a fraud (*Dias, 2021*). The Association of Certified Fraud Examiners (ACFE) classifies occupational fraud into three types, asset misappropriation, corruption, and fraudulent statements. Asset misappropriation refers to the theft or misuse of an organization's assets. Corruption influences a business transaction for personal gain, and misrepresentation is the intentional misrepresentation of financial or non-financial information to deceive others (*Shruti, 2018*). Several theories allow analyzing the problems related to fraud, which serve as a guide for organizations to combat this phenomenon, contributing to the prevention, detection, and deterrence of activities related to occupational fraud. Why is labor fraud committed within organizations? This question explains the fraud triangle, the first model developed to address this problem. This theory has been the basis for creating tools to deal with this crime. However, it has its limitations, which do not cover all fraud cases due to the progress and sophistication of this behavior, so developing a model that includes all fraud cases is a challenge (*Moore, 2020*; *Huang et al., 2017*). Over the past 60 years, the fraud triangle has evolved into various models, including the diamond and the fraud pentagon. The FTT was proposed by *Machado & Gartner (2017)*; FTT identifies three crucial elements: pressure, opportunity, and rationalization. According to this theory, fraud is typically accompanied by pressures/incentives, opportunities, and rationalizations/attitudes. Thus, it is highly probable that the perpetrator is driven by pressure or motivation to commit fraud. Additionally, the perpetrator will likely find potential opportunities to carry out their fraudulent actions. Moreover, they can rationalize their deceitful acts by justifying their necessity. Ultimately, all three conditions directly correlate with a heightened likelihood of fraud (*Puspasari, 2015*). This theory later evolved into the fraud diamond theory by adding a new element, capacity, proposed by *Wolfe & Hermanson (2004)*. Finally, the Pentagon theory of fraud is the latest evolution proposed by *Marks & Ugo (2012)*, to which two elements were added: competition and arrogance. Competition in this model has the same meaning as the ability described by *Wolfe & Hermanson (2004)*, aiming to perfect the diamond theory of fraud (*Hidayah & Saptarini, 2019*). The elements or variables associated with the different fraud theories are directly related to the behavior of the perpetrators, which are clear indicators that can cause fraud. The triangle, diamond, and pentagon of fraud are relevant theories that can be used interchangeably effectively to detect the possibility of fraud, depending on the existence and availability of evidence related to the variables of these theories (*Christian, Basri & Arafah, 2019*). The effectiveness of the fraud triangle theory has been proven in *Muhsin & Nurkhin (2018)*, evidencing more precise results on the fraud diamond and pentagon because the capacity and arrogance of the variable in many cases do not significantly affect the behavior of fraudsters. Individuals will not commit fraud despite great ability and arrogance. In this context and because the characteristics of the study dataset are aligned with the triangle theory of fraud, this model will be used to develop this work.

## Topic modeling

Topic modeling (TM) is a statistical technique that has revolutionized text mining, allowing the discovery of semantic structures in a collection of documents (*Vayansky & Kumar,*

*2020*). Popular algorithms for multidomain text analysis include latent semantic analysis (LSA), non-negative matrix factorization (NMF), probabilistic latent semantic analysis (PLSA), and latent Dirichlet allocation (LDA). LSA and NMF work on a bag-of-words (BoW) model-oriented approach, a text representation describing the occurrence of words within a document, which converts a corpus into an array of document terms. On the other hand, LDA and PLSA were initially unsupervised approaches, which evolved into supervised and semi-supervised models, respectively (*Kherwa & Bansal, 2018*). These models have weaknesses associated with their design; in the case of LSA, obtaining and determining the optimal number of topics is a complex task. PLSA has several overfitting problems, and LDA often does not expose the relationships between topics. To circumvent these difficulties, topic modeling with a semi-supervised approach allows previous knowledge to be provided in the topic model. Specifically, there are versions in which the model can be given "seed" words of the study topic, and the model's algorithm encourages topics to be built around these seed words; this solves the problems mentioned above and allows us to direct topics toward relevant topics simply by adding keywords while leaving room for discovering "unknown" topics. In this context, alternative models to the traditional techniques have been developed in the semi-supervised approach, such as Correlation Explanation (CorEx), which, unlike LDA, does not make assumptions about the data generation process but instead addresses the modeling of issues.

In an information-theoretic way, they avoid time and effort to identify topics and their structure ahead of time. On the other hand, guided LDA (GuidedLDA), a variant of LDA, improves the performance of topics that infrequently occur, where a variation of the LDA algorithm is made so that the topic-word and topic-document distributions take into account the seed words (*Toubia et al., 2019*). They have also appreciated techniques such as the Dirichlet multinomial mixture (DMM) that allows overcoming data scarcity problems in short texts, generally below 500 characters (*Mehrotra et al., 2013*).

### GuidedLDA

GuidedLDA or SeededLDA implements LDA and can be guided by setting some seed words per topic, which will cause topics to converge in that direction (*Singh, 2022a*). In the study by *Andrzejewski, Zhu & Craven (2009)*, they used words that belong to specific topics and are limited to appearing in some subset of all possible topics. A second model proposed by *Andrzejewski & Zhu (2009)* uses relationships between words to break up confusing topics. While in *Jagarlamudi, Daumé III & Udupa (2012)*, they propose SeededLDA, an extension of semi-supervised LDA, and use seed words to influence both topic-word distributions and document-topic distributions; it is a model that guides but does not force these topics into seed words. Specifically, the generative process of estimating these distributions is guided by initial word-level information, a set of user-defined words characteristic of the topics in the study corpus. This approach allows the user to provide N sets of representative seed words from the corpus to guide the topic discovery process. These "seed sets" correspond to the word sets preliminarily obtained by the LDA (*Zhou et al., 2023*) model. To obtain contextually relevant topics, such as the impacts of fraud, strategies, and initiatives for its prevention and mitigation; some initial keywords highlighted by topic must be established,

allowing us to obtain topics that help us understand the content of the dataset we are analyzing.

### (Anchored) correlation explanation

CorEx is a topic model based on total correlation explanation, which allows identifying topics in a corpus and explaining their structure through the dependency found on the data (*Steeg & Galstyan, 2014*). In addition, it is a seeded technique with several advantages over the seeded LDA variant (SeededLDA), such as a better consistency of the derived topics and good algorithmic performance (*Gallagher et al., 2017*). Unlike LDA, CorEx makes no assumptions about the data generation process but instead approaches topic modeling in an information-theoretic way. This model allows the incorporation of domain knowledge through user-specific anchor words that guide the model to topics of interest; this allows the model to represent topics that do not arise naturally and provides the ability to separate keywords that allow topics to be identified differently (*Sockin, 2022*). Anchored CorEx optimizes the following in Eq. (1) (*John et al., 2019*):

$$\underset{X;Y}{\text{Maximize}} \ TC(X;Y) + \beta \sum I(x;y) \tag{1}$$

where $X$ and $Y$ are random variables, $TC$ and $I$ represent the total correlation and mutual information, respectively, and $x$ is an anchor word.

## Classification methods

Classification problems have been deeply analyzed and have aroused the scientific community's interest, mainly applied in data analysis in machine learning, statistical inference, and data mining (*Novaković et al., 2017*; *Gaber, Zaslavsky & Krishnaswamy, 2007*). In general, classification is a data mining approach used to predict the membership of a data instance to a given class from a set of predefined classes (*Soofi & Awan, 2017*; *Sarker, Kayes & Watters, 2019*).

Given such a diversity of methods, the question arises as to which method should be used for a problem to be solved. The answer depends on the nature and approach with which the problem is addressed. So there will be many performance measures, each addressing different aspects (*Hand, 2012*).

Frequently, the performance of a combination of indicators is quantified by indices related to the receiver operating characteristic (ROC) curve: sensitivity, specificity, or the area under the curve (AUC) (*Yu & Park, 2014*). A ROC curve is a graph that shows the relationship between the true positive rate (TPR, or specificity) on the $y$-axis and the false positive rate (FPR, or 1–specificity) on the $x$-axis (*Wu et al., 2008*). The ROC curve shows the performance of a classifier without considering the class distribution or the cost of misclassification. The area under the receiver operating characteristic curve must be determined to compare the ROC curves of various classifiers (*Brown & Mues, 2012*). The area under the ROC curve, or AUC, measures model performance for all possible decision thresholds. It gauges the overall performance of a test set and is interpreted as the average sensitivity value for all potential specificity values. Since the $x$ and $y$ axes have 0 to 1, it can take any value between 0 and 1 (*Park, Goo & Jo, 2004*).

**Table 2 Description of two classification methods: random forest (RF) and gradient boosting decision tree (GBDT).**

| Model | Description | References |
|---|---|---|
| Random Forest | A tree-based ensemble where a set of random variables determines each tree. Decision trees are chosen randomly from the available data, and the averaging process helps mitigate low bias and high variance. | *Zhang & Ma (2012)*, *Louppe (2014)*, *Ali et al. (2012)* |
| Gradient Boosting Decision Tree | Use decision trees as weak classifiers for regression or classification tasks with logarithmic loss. It combines the results of multiple variables sequentially to outperform earlier outcomes by using gradient increase to train predictors and repair previous mistakes. | *Deng et al. (2021)*, *Alcolea & Resano (2021)*, *Chong, Xinrui & Zipei (2020)* |

Using the AUC criterion, this paper compares random forest (RF) and gradient boosting (GB) methods to detect fraud-related text. A description of these algorithms can be seen in Table 2.

## METHODOLOGY FOR PREDICTING FRAUD BASED ON THE FRAUD TRIANGLE COMPONENTS

Implementing a predictive model that identifies hidden patterns related to suspected fraud is the objective of this work, for which topic modeling was used and, specifically, the most relevant techniques used in text mining, such as LSA, NMF, and LDA, were tested. A comparison was made to identify these algorithms' efficiency and determined that LDA is the most consistent model. To determine the number of topics, the coherence value or parameter k was used as a metric, which allows us to identify the most appropriate number of topics of the three models that adjusts to the nature of the information used and indicates the level of similarity. Semantics exist between words for each topic. LDA allows finding topics to which a document belongs based on the words it contains; This served as a starting point to identify the most representative words and their distribution in the different topics. This initial strategy served as a starting point for using semi-supervised learning algorithms by using some initial words for the topics considered most representative of the underlying themes in the study corpus. It guided the models to converge around those terms. This way, we observe how the models can configure the seed words to guide their results in a particular direction.

The application of topic modeling aims to determine the probability that a document within the study corpus belongs to a specific topic that aligns with the vertices of the fraud triangle. This crucial step, depicted in the first phase of Fig. 1, identifies potential fraud-related behaviors. These probabilities are then used to train various classification methods, allowing for predicting suspicious activity associated with fraud. Evaluating the performance of the different classifiers is essential in selecting the one most compatible with the topic analysis carried out for fraud detection. This fundamental evaluation stage, illustrated in the second phase of Fig. 1, ensures the effectiveness and accuracy of the chosen classifier.

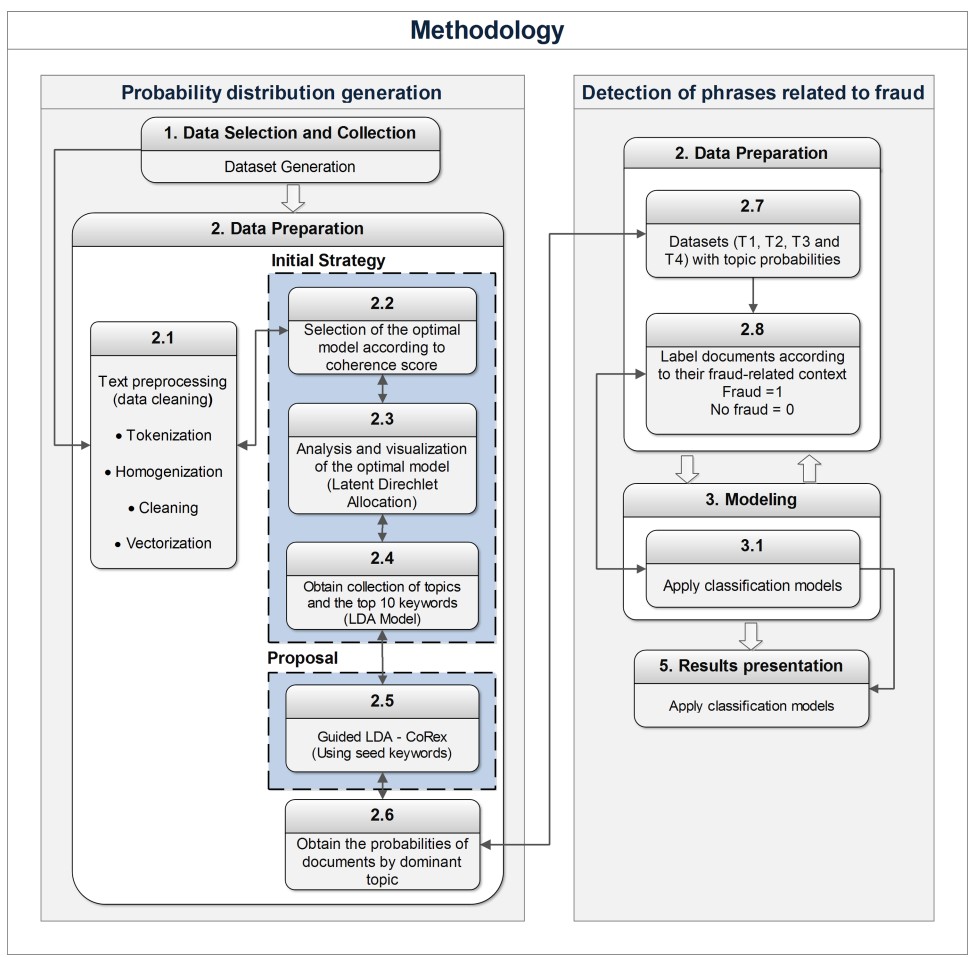

**Figure 1** **Methodology used to determine the existence of fraud.**

## Dataset generation

One of the most difficult challenges in the analysis and study of fraud is the lack of information related to this phenomenon. The datasets that contain evidence that identifies fraudulent activities or suspicions of possible commissions of this crime are scarce and difficult to access due to their confidentiality, rights, and intellectual property. Due to these difficulties, a practical solution that solves this need is to generate synthetic data, which becomes a viable strategy for studying this phenomenon. According to several studies, this data type allows experimentation using Machine learning techniques to be faster and more efficient, producing data with similar characteristics to reserved and difficult-to-access information (*Guan et al., 2019*).

In *Sánchez-Aguayo, Urquiza-Aguiar & Estrada-Jiménez (2022)*, they used a synthetic dataset generated from a dictionary of keywords related to the fraud triangle, which we will call WebScraping. Through the use of online tools (*RandomWordGenerator, 2022b*) that allow the production of grammatically well-defined sentences from the entry of specific words, which for this particular case were used those belonging to the dictionary of

words related to fraud and its vertices (pressure, opportunity, and rationalization), which served to build the study dataset in this research. This study proposed an initial fraud prediction model supported by the synthetically generated dataset. To validate the usability of synthetically generated data, *Sánchez-Aguayo, Urquiza-Aguiar & Estrada-Jiménez (2022)* compares and evaluates the performance of different synthetic datasets (WebScraping and Neural-Network) *versus* an original (Students) to demonstrate whether the synthetic data can be used as a substitute for real data, noting that according to the performance metrics obtained from the comparison made, this alternative to real data is reliable and serves as a valid option for data analysis. The experimentation on the real and synthetic datasets allows the identification of similar behaviors in the results based on their performance after applying the fraud prediction method, which suggests that the different datasets analyzed can be generalized to different scenarios. In this context, the datasets mentioned above were used for the present work, with the objective of contrasting results after applying modeling of semi-supervised topics and classification algorithms *vs.* the first fraud prediction model proposed.

A dataset called ChatGPT was also generated, with the same characteristics as the previous ones, which this tool was used to build. This artificial intelligence made it possible to generate phrases related to the three vertices of the fraud triangle, for which messages were entered that consisted of imagining scenarios that include the elements of the theory; in the case of opportunity, they were told to "imagine a scenario in which that an employee has access to confidential financial information," to incorporate the element of rationalization, he was asked to "imagine a scenario in which an employee justifies his fraudulent actions by believing that he is not being fairly compensated" and finally, to incorporate the pressure element, "Imagine a scenario where an employee experiences financial hardship that motivates them to commit fraud." Once this environment was created, the generation of a certain number of sentences oriented to each vertex of the triangle was promptly requested, entering messages requesting "Tell me sentences that a person can say regarding pressure, opportunity, or perceived rationalization."

## Data preprocessing

Within Artificial Intelligence, natural language processing (NLP) is the field that is responsible for investigating how computers understand, analyze, and interpret human language. NLP allows people to interact with machines in human language. Computer languages only work correctly when correctly written because they are precise in their syntax. At the same time, the flexibility of natural languages allows them to adapt to their nature and interpret errors such as accents, words, and dialects (*Kirwan & Zhiyong, 2020*). One of the most common NLP tasks is to clean up text data. Extracting the text to the most critical root words in the corpus maximizes the results. Text preprocessing in NLP is a method that allows cleaning up text so that it is ready to feed models. Noise in the text comes in various forms, like punctuation and different cases. All these noises are not helpful for the machines and therefore need to be cleaned (*Hegazi et al., 2021*). In text mining, preprocessing involves a 3-step mechanism that includes extraction, stopword removal, and lemmatization. Extraction is the process of breaking down documents into individual

elements, forming a format composed of tokens, words, terms, or attributes. These features represent the document in a vector space, with their weights determined by the frequency in the text document. Removing stopwords, numbers, and special characters helps reduce the dimensionality of the term space. Lastly, lemmatization standardizes words by reducing them to their etymological root, eliminating common suffixes, and reducing word count (*Kadhim, 2018*).

## Quantitative evaluation of topic modeling algorithms

The structured grid search technique was used to identify the optimal topic modeling. The results obtained by the unsupervised LDA algorithm were compared with semi-supervised models such as CorEx and guided LDA. In related works, comparisons between these techniques are presented, evidencing, in most cases, the superior performance of semi-supervised algorithms over unsupervised ones (*Alnusyan et al., 2020*; *Egger & Yu, 2021*). In this context, there is little evidence of studies comparing semi-supervised models. It is necessary to analyze the efficiency of these semi-supervised algorithms, for which, in the first instance, the same text preprocessing techniques and the hyperparameters used as input for the models were used, and they were evaluated through the coherence value "C_v." That determines the performance of the algorithms the different topic modeling algorithms. This metric allows us to identify how coherent a model is about the structure of its topics; the more different the words are in each topic, the less related the topics will be, and the more coherent the model will be. Once the hyperparameter k or an adequate number of topics was identified to obtain adequate modeling, the models were tested using seed word dictionaries to more easily generate topics corresponding to the identified categories.

## Selection of the topic modeling algorithm

Once the models generated by GuidedLDA and CorEx were obtained, the consistency of the sets of words per topic formed was analyzed. The efficiency of the semi-supervised algorithms to establish the distributions in each topic was determined. This analysis allows us to identify, according to the parameter k, the most suitable method to distribute the words in their respective topics more efficiently. With the different modeling results, we analyze the word distributions by topic and identify the words related to fraud and their behavior within the distribution. We point out those that coincide with the seed word dictionaries used, which are associated with the vertices of the fraud triangle. The objective is to show if the topics generated are associated or related to the vertices of the pressure, opportunity, and rationalization triangle. After this analysis, we select the model with the best performance and that most consistently brings together the words related to fraud by topic, through which we will obtain the probabilities that the documents in the study dataset belong to a given topic. The different probabilities obtained represent a measure that makes it possible to identify whether a document is related to fraud and express a new representation of the dataset. Then, we build smaller datasets from this new dataset, each of which groups documents associated with a "dominant" topic, a topic to which the documents most likely belong.

## Evaluation

Once the appropriate topic modeling for the present case study has been identified and the probability distributions of documents per topic have been generated, it is feasible to use Machine learning techniques to predict fraud-related activities. When small datasets are available, traditional classifiers frequently learn better than deep learning classifiers, which gives us a guideline for selecting the appropriate techniques. The graph of the ROC curve and its area under the AUC curve was used to evaluate the performance and identify how accurate the prediction of the classification methods used in the experiment, which represents the quality of the methods, which allows us to visualize the behavior of each of these and analyze their performance.

In addition, as part of the evaluation process, training on the proposed model will be carried out using datasets generated one at a time. Later, it will be tested with the remaining datasets; this will allow for obtaining more accurate and reliable results on the effectiveness and performance of the model in different scenarios, guaranteeing a comprehensive evaluation of its performance under different conditions.

# RESULTS AND DISCUSSION

This section presents the results obtained from testing our improved fraud detection model. The efficiency of the results is analyzed and discussed in this section. Details about the experimentation, from selecting topic modeling to applying machine learning models, are reviewed. Finally, the different theories, techniques, and models applied to the approach of this model are discussed.

## Probability distribution generation

The first stage of the experimentation consists of applying topic modeling techniques to the study dataset to identify hidden patterns related or not to fraud and analyze how consistent these results are; This is to obtain information structured by topics that, once the model is applied, allows us to analyze its characteristics based on the probability that a document belongs to a specific topic.

### *Initial strategy—application of the LDA model*

Of the topic modeling algorithms reviewed in *Sánchez-Aguayo, Urquiza-Aguiar & Estrada-Jiménez (2022)*, it was determined that LDA has the best behavior when analyzing data related to fraud since it more consistently groups words by topic. After carrying out different tests in the experimentation, it was validated that the adequate number of topics is four. With this value, the LDA algorithm is applied to the study dataset, obtaining a distribution of words categorized into four topics according to their context and problem of study. The present work relates to the vertices of the FTT, "pressure, opportunity, and rationalization," and another topic they call others. This distribution of words can be seen in Table 3, which is ordered by topic and prevalence. In addition, the words related to fraud are formatted to identify that they belong to a specific vertex of the fraud triangle. Words unrelated to fraud that belong to said vertices were not formatted. The top 20 terms are manually analyzed and filtered to use only the most significant ones (for each topic).

**Table 3 The most frequent terms in the dataset connected to each of the three vertices of the fraud triangle are found after LDA has been applied.** To represent the vertices of pressure, rationalization, and opportunity, the words are represented by bold, underlined and italics.

| Topics | | | |
|---|---|---|---|
| **T1** | **T2** | **T3** | **T4** |
| **review** | **debt** | **problem** | want |
| care | think | **economic** | know |
| poor | later | make | **job** |
| steal | fix | big | work |
| temporary | just | people | **lose** |
| say | tell | abuse | *support* |
| new | *inadequate* | fair | **deadline** |
| man | look | compensation | help |
| really | *failure* | child | come |
| *insufficient* | *weakness* | good | time |
| state | ill | *earning* | **exploitation** |
| money | unethical | *easily* | deserve |
| issue | life | accessible | **scare** |
| *evacuation* | world | country | right |
| leave | try | need | like |
| woman | let | way | day |
| year | talk | pay | use |
| long | old | school | scared |
| change | feel | home | ask |
| period | place | thing | car |

Table 3 presents inductive labels, presenting the main terms identified by the class and the most significant to be used as seed or anchor words for the semi-supervised models, which are identified by a text format. For example, in Topic 2 (T2), words like "life" or "word" are related to a different approach to fraud, and, therefore, we did not choose them as meaningful representations for one of the vertices of the fraud triangle. As can be seen, the words related to fraud are distributed through the topics without distinguishing groups in the different topics; this indicates that the topics obtained through LDA cannot be directly associated with the vertices of the fraud triangle. However, due to the presence of a high number of words with a high degree of repetition in the different topics, the existence of behavior related to fraud follows.

### Proposal—explore topic modeling using semi-supervised learning

Classical topic modeling methods are algorithms that generate various topics from a study dataset. However, due to their unsupervised nature, these methods can impede the comprehension of the analyzed texts. They are prone to create less essential topics, leaving aside several others that may interest them *Steeg (2017)*. Each word is randomly assigned to a topic in LDA, controlled by Dirichlet priorities through the Alpha parameter. Then it is required to find out which term belongs to which topic. LDA uses a straightforward approach to finding the topic for one term at a time. Suppose we want to find the topic of

the word "problem" related to fraud. The algorithm distributes each word evenly across all the topics found and assumes it is the right topic for those words. Then, find out what other words the word "problem" is associated with most often. In this context, it is determined what the most common topic among those terms is. Therefore, the word "problem" is assigned to that topic. The word "problem" is close to any topic where words like "debt" and "need' are found. These three words are closer to each other before this step. Finally, the model moves on to the next word and repeats the process as many times as necessary to converge. Semi-supervised topic modeling allows the introduction of prior knowledge by incorporating words called "seed" or "anchor" into the algorithm that stimulates or encourages (but does not force) the model to build topics around these anchor words. This alternative of adding keywords gives the flexibility to generate relevant topics while allowing the discovery of unknown topics. GuidedLDA and CorEx use tag seed words to make their training converge around these words. That is, a set of specific words relevant to a tag related to the same topic is used, and the weight of these particular words is increased during training to capture other strongly related words. In other words, these seed words function as anchors.

The coherence score aims to measure the similarity between words and how interpretable the topics obtained by the model are. Starting from the premise that we have the reference coherence score obtained for the LDA model, in which several sensitivity tests were carried out to determine the adequate number of themes, they established C_v as a metric for performance comparison. Since the coherence score gradually increased with the number of topics, the model with the highest C_v was chosen. In this case, $K = 4$. For the semi-supervised models that we will use in this proposal and considering that we will use the same study corpus, we will use this value of K-topics to perform the respective tests.

As mentioned, semi-supervised topic models require a list or set of keywords called seed or anchor related to each topic for modeling. These words are used to identify specific topics; in this sense, they are related to the three vertices of the FTT for the present work. In these models, a force or push parameter defines the bias of the generated topics toward the seed or anchor keywords. This value can vary between models; for the case of GuidedLDA, it can range between 0 and 1. A 0.1 can bias the seed words by 10% more toward the seed topics. On the other hand, in CorEx, it should always be above 1, and higher values indicate a more substantial bias toward anchor keywords. In this context, the list of anchor keywords for the models was provided, those that were generated in the initial strategy applying LDA, represented in Table 3, and those words with the greatest representativeness related to the three vertices were chosen from the different topics "pressure, opportunity, and rationalization."

Words are initialized by setting tags as keys and a list of initial words (relative to critical topics) as values.

Table 4 shows the results of applying GuidedLDA and CorEx topic modeling and the top 20 terms identified by topic. It can be observed how the words of the study corpus are distributed in the four established topics. These words are organized by topic and prevalence to identify those that the model considers most relevant. Using the same procedure as *Sánchez-Aguayo, Urquiza-Aguiar & Estrada-Jiménez (2022)*, we format the

**Table 4  The terms that appear most frequently in the study dataset are associated with each of the three vertices of the fraud triangle once GuidedLDA and CorEx have been applied.** The words are represented by bold, underlined and italics to indicate the vertices of pressure, rationalization and opportunity, respectively. CorEx better classifies the terms by topic.

| Topics | | | | | | | |
|---|---|---|---|---|---|---|---|
| **T1** | | **T2** | | **T3** | | **T4** | |
| G-Lda | CorEx | G-Lda | CorEx | G-Lda | CorEx | G-Lda | CorEx |
| **review** | **problem** | time | *support* | people | care | think | people |
| **debt** | **economic** | system | *failure* | study | poor | write | think |
| **economic** | **review** | love | *easily* | think | deserve | lose | time |
| study | **job** | *failure* | *insufficient* | play | later | people | love |
| **problem** | **lose** | study | *inadequate* | abuse | compensation | nobody | play |
| **deadline** | **deadline** | write | *evacuation* | **economic** | fix | care | privacy |
| *earnings* | **exploitation** | play | *earning* | accessible | steal | deserve | tank |
| compensation | labor | *error* | *supervision* | poor | temporary | steal | song |
| *inadequate* | period | fix | *error* | **exploitation** | fair | **job** | album |
| fair | currently | *weakness* | accessible | **problem** | unethical | play | update |
| *insufficient* | solve | *evacuation* | security | many | illegal | poor | indigenous |
| **problems** | social | accessible | muscle | unethical | trade | fix | spend |
| play | political | think | file | *supervision* | seek | something | change |
| *supervision* | issue | temporary | datum | temporary | know | unethical | live |
| **exploitation** | country | **job** | strength | role | alcohol | want | make |
| countries | work | file | remain | **problems** | victim | song | people |
| role | external | use | capacity | children | try | things | think |
| period | face | case | warn | love | verbal | good | time |
| people | | data | | *weakness* | | anything | |
| *evacuation* | | change | | work | | look | |

words to identify their belonging to each vertex of the fraud triangle. Those not related to the vertices were not formatted. We can observe that the model obtained by GuidedLDA has a behavior similar to that of regular LDA, which does not reflect a relationship between the topics obtained with each of the vertices of the fraud triangle since the words within each topic contain words associated with different vertices of the triangle. As in regular LDA, the model does not group words into topics related to each vertex of the fraud triangle. Still, the probability that the corpus documents belong to each topic provided by the model is helpful for feed classification algorithms to detect whether or not a phrase is related to fraud. On the other hand, the results obtained by CorEx are more interpretable than their predecessor since each topic obtained can be linked directly with the knowledge of the domain established in the list of initial words or anchors. A clear relationship can be seen between the resulting topics and the vertices of the fraud triangle; for example, in topic 1 (T1-CorEx), the words (bold) related to pressure are grouped in order of importance; in topic 2 (T2-CorEx), the words (italic) related with opportunity, topic 3 (T3-CorEx) the words (underline) related to rationalization and finally topic four those that are not related to fraud. CorEx allows the obtained model to converge to link the seed or anchor words to a given topic.

**Table 5  Probabilities obtained from GuidedLDA (G-LDA) and CorEx in the different established topics; where each row represents a specific result for a particular model, the values in the G-LDA column represent the probability obtained by this model that a document belongs to that topic.** In contrast, the values in the CorEx column have binary values, where true indicates that the document belongs to that category, and false indicates what is contrary.

| Docs | Pressure | | Opportunity | | Rationalization | | Others | |
|------|------|------|------|------|------|------|------|------|
| | G-LDA | CorEx | G-LDA | CorEx | G-LDA | CorEx | G-LDA | CorEx |
| 0 | 0.43 | False | 0.12 | True | 0.45 | False | 0.00 | False |
| 1 | 1.00 | False | 0.00 | False | 0.00 | False | 0.00 | False |
| 2 | 1.00 | False | 0.00 | False | 0.00 | False | 0.00 | False |
| 3 | 0.01 | False | 0.00 | True | 0.98 | False | 0.01 | False |
| 4 | 0.00 | False | 0.00 | False | 0.23 | False | 0.77 | True |
| 5 | 0.99 | False | 0.00 | True | 0.00 | True | 0.01 | False |
| 6 | 1.00 | True | 0.00 | False | 0.00 | False | 0.00 | False |
| 7 | 0.23 | True | 0.00 | False | 0.00 | False | 0.77 | False |
| 8 | 0.99 | False | 0.00 | True | 0.01 | False | 0.00 | False |
| 9 | 1.00 | False | 0.00 | True | 0.00 | False | 0.00 | False |

keywords = [

[‘economic’, ‘problem’, ‘deadline’, ‘review’, ‘debt’, ‘exploitation’,
‘lose’, ‘job’, ‘scared’],

[‘earning’, ‘insufficient’, ‘inadequate’, ‘evacuation’, ‘supervision’, ‘weakness’,
‘error’, ‘failure’, ‘support’, ‘easily’],

[‘deserve’, ‘abuse’, ‘fair’, ‘temporary’, ‘unethical’, ‘poor’,
‘steal’, ‘care’, ‘fix’, ‘later’],

[‘love’, ‘study’, ‘think’, ‘time’, ‘people’, ‘write’, ‘play’, ‘game’, ‘passion’]

]

Once the models are obtained, the four resulting topics are manually labeled concerning the three vertices of the fraud triangle fraud pressure, opportunity, rationalization, and others. This categorization of topics is essential since it allows for interpreting the corpus and identifying the implicit topics in a dataset. The interpretation of a topic can be achieved by examining a ranked list of terms in each topic (*Merino & Atzmueller, 2019*).

With the defined models, we can obtain the probability that a particular document in our corpus belongs to a specific topic; by entering the document into the model, it analyzes it and calculates the possibility of belonging to each one of the topics, establishing a percentage of probability per topic. One approach to classifying a document as belonging to a particular topic is to analyze which topic contributed the most to that document and assign it to that topic. Table 5 shows the percentages of belonging to a document associated with each topic. In this case, the one with the highest value corresponds to the most related or dominant for GuidedLDA.

Applying the same procedure to CorEx, it can be seen in Table 5 that the algorithm returns boolean values (True or False) to determine if a topic contributed more to a document and if this document is more related to that topic. In general, this approach

could inform about a document belonging to a specific topic without specifying the weights to which each topic contributed to that document.

Given that the metrics corresponding to the probabilities obtained by the models are necessary to feed classification models and their subsequent fraud prediction, it is essential to obtain the required values and be able to process them using machine learning algorithms.

In this context, the operation of the "corextopic.py" module, developed in Python, which contains the functions associated with transforming the data according to the previously defined model, was analyzed. The transform() algorithm ?? takes a matrix X consisting of (n_samples, n_visible), where n_samples is the number of data points, and n_visible is the dimensionality of each data point. The input data samples are preprocessed by applying a normalization or standardization, which is done by calling the preprocessing method; the preprocessed data is then stored in X. The latent_calculation method is then called to calculate the latent variables p(y—x) and the log-likelihood of the data log(z) for the preprocessed data samples X and the model parameters (self.theta). The resulting values are stored in p_y_give_x and log_z, respectively. Finally, the label() method is called to assign a label to each data sample; this algorithm ?? is inside the same "corextopic.py" class, which takes the matrix p_y_given_x of the form [n_samples, n_hidden] that represents the distribution over the hidden variables given the observed variables and returns binary labels for each sample based on the estimate of maximum likelihood. Additionally, it applies a threshold of 0.5 to the probabilities at p_y_given_x. If the probability of the hidden variable is more significant than 0.5, the method assigns it a true label; otherwise, it assigns a false label. The output is a boolean array of the form [n_samples, n_hidden] representing the labels of each sample. The resulting labels are stored in the labels variable.

---

**Algorithm 1:** Label hidden factors for (possibly previously unseen) data samples.

---

**Input:** $samples of data, X, shape = [n\_samples, \_visible]$     // List of Sensitive Terms

**Output:** $shape = [n\_samples, n\_hidden]$          // Negation Excluded List

1 **Function** transform(*Takes in two inputs: X, and details*)**:**

2     $X$ ← **self.preprocess((X))** $p\_y\_given\_x, \_, log\_z$ ← **self.calculate_latent((X, self.theta))** *labels* ← **self.label((p_y_given_x))**

3     **if** details is true: **then**

4        **return** *return p_y_given_x, log_z*

5     **end**

6     **return** *labels*

7 **End Function**

8 **Function** label($p\_y\_given\_x$)**:**

9     **return** $(p\_y\_given\_x > 0.5).astype(bool)$

10 **End Function**

---

Because it is required that the estimation of the document-topic distributions be obtained as a return value, it is necessary to update the "transform" method, for which its counterpart of the GuidedLDA algorithm was taken as a reference in the "guidedlda.py" module, this method applies topic modeling using latent Dirichlet assignment (LDA) on a document term matrix X. In this case, the transform() algorithm ?? takes the document term matrix X as a numpy array and the parameters "max_iter" and "tol" to control the convergence of the model. Stores the topic distribution for each document in the corresponding row of the doc_topic array and returns this array containing the probability values corresponding to the topic distribution for each document.

---

**Algorithm 2:** Transform the data X according to previously fitted model

**Input:** $X : array - like, shape(n\_samples, n\_features), max\_iter : int, optional, tol : double, optional$

**Output:** $doc\_topic : array - like, shape(n\_samples, n\_topics)$   // Point estimate of the document-topic distributions.

```
 1  Function transform(self, X, max_iter = 20, tol = 1e − 16):
 2      if isinstance((X, np.ndarray)) then
 3          X ← np.atleast_2d((X))
 4      end
 5      doc_topic ← (np.empty)((X.shape[0],self.n_hidden)) WS, DS ←
          lda.utils.matrix_to_list((X))
 6      foreach d ∈ np.unique((DS)) do
 7          doc_topic[d] ← self._transform_single((WS[DS == d], max_iter, tol))
 8      end
 9      return doc_topic
10  End Function
```

---

Once the changes have been made, the module is imported again. It generates the results with the probabilities by topic and dominant topic, as seen in Table 6.

In addition to the probabilities obtained, we label the first 7,113 records with 1 to indicate that these documents are fraud-related and the remaining 7,113 with 0 to indicate otherwise. A filter by dominant topic is applied to this new representation of the dataset 6, obtaining four datasets per topic (pressure, opportunity, rationalization, and others) that served as input to train classification algorithms.

### Detection of phrases related to fraud

The second stage of the experimentation consists of applying classification methods to the datasets (pressure, opportunity, rationalization, and others) with the probabilities obtained in the previous stage through the application of the semi-supervised algorithms (GuidedLDA and CorEx). To try to predict behaviors suspected of fraud.

Once the dominant topic filters the original dataset, four datasets are generated, labeled as fraud and non-fraud for all their records. We build models using these new representations and classification algorithms to predict whether a new document inputted into the model

**Table 6  Numerical representation of the distribution of probabilities by topic (pressure, opportunity, rationalization, and others) obtained through CorEx modifying the transform() method.** To the 14,229 documents that comprise the corpus, an additional column is added that identifies the dominant topic (DT), representing the highest probability that a document belongs to a specific topic.

| Docs | Pressure | Opportunity | Rationalization | Others | DT |
|------|----------|-------------|-----------------|--------|-----|
| Doc 0 | 0.02 | 0.82 | 0.16 | 0.00 | 1 |
| Doc 1 | 0.13 | 0.36 | 0.02 | 0.49 | 3 |
| Doc 2 | 0.25 | 0.65 | 0.10 | 0.01 | 1 |
| Doc 3 | 0.01 | 0.66 | 0.34 | 0.00 | 1 |
| Doc 4 | 0.00 | 0.34 | 0.04 | 0.62 | 3 |
| ... | ... | ... | ... | ... | ... |
| Doc 14225 | 0.25 | 0.07 | 0.00 | 0.68 | 3 |
| Doc 14226 | 0.00 | 0.20 | 0.42 | 0.38 | 2 |
| Doc 14227 | 0.00 | 0.00 | 0.30 | 0.70 | 3 |
| Doc 14228 | 0.00 | 0.06 | 0.00 | 0.94 | 3 |
| Doc 14229 | 0.00 | 0.73 | 0.02 | 0.26 | 1 |

is related to fraud. RF and GB algorithms were applied due to their superior performance, as reported in *Sánchez-Aguayo, Urquiza-Aguiar & Estrada-Jiménez (2022)*.

### Classifier performance

In the present work, the ROC curve was used to represent the performance of different machine learning models when classifying documents as related or unrelated to fraud. Several metrics, including recall, accuracy, and precision, can be used to assess the performance of a classification model. One of the main weaknesses of these metrics is that they are susceptible to changes in class distribution. When the ratio of positive to negative occurrences in a test set changes, a model's performance may no longer be optimal or acceptable. However, the ROC curve is independent of the class distribution changes (*Wu, Flach & Ferri, 2007*), so for this type of analysis, it is a frequently used technique (*Trifonova, Lokhov & Archakov, 2014*; *Mallett et al., 2014*). The ROC curve will not change even if there is a change in the class distribution of a test set; This is because the ROC curve is based on the underlying class conditional distributions from which the data is drawn. It plots a model's true positive rate on the $y$-axis against its false positive rate on the $x$-axis. It provides a general measure of model performance, regardless of the various thresholds used. The results can be seen in Fig. 2 but are also presented in Table 7.

Based on these findings, random forest and gradient boosting perform the best, with a mean area under the curve (AUC) of 0.87 and 0.88, respectively. These findings imply that our method to identify fraudulent actions based on topic identification using semi-supervised models would be feasible when developing machine learning models.

### Comparison of classification algorithms

When comparing the performance of the classification methods, it was observed that RF and GB showed similar performances, with an average AUC of 87% and 88%, respectively, as shown in Table 7. Furthermore, GB exhibited a slight superiority of 1% about RF. These

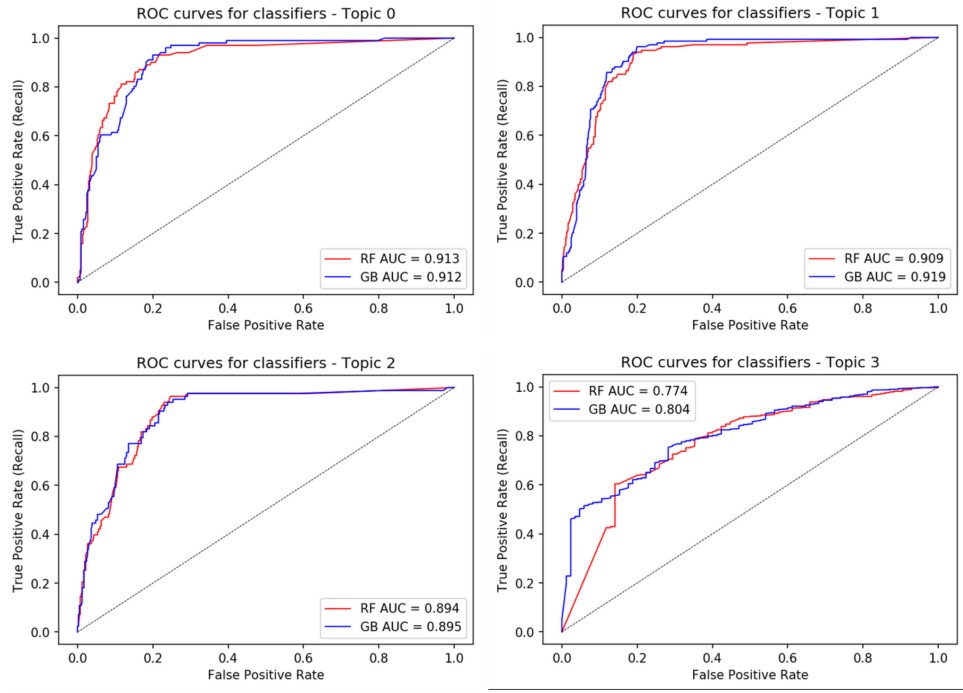

**Figure 2    The ROC curves of different machine learning classification models.** The models are: RF and GB. The results show that GB obtained the highest AUC in all the topics.

findings align with the results reported in *Sánchez-Aguayo, Urquiza-Aguiar & Estrada-Jiménez (2022)*, where RF and GB were identified as the most efficient classification algorithms, achieving an average AUC of 81%. By using semi-supervised methods for topic modeling, a notable improvement of 7% was observed in the performance of the classification methods to predict behaviors suspected of fraud; this suggests that incorporating the semi-supervised approach improves obtaining document probabilities by topic, increasing the accuracy and efficiency of fraud prediction models that use RF and GB algorithms.

## Validation

The validation of a model consists of evaluating its performance using a dataset that has yet to be used during the training process. The main goal of validation is to estimate a model's performance and get an idea of how well it will work with new data. When building a machine learning model, it is necessary to guarantee its performance through a proper validation process. A standard model validation method uses learning curves and graphs showing the relationship between model performance on training and validation sets as a function of the training data. Observing the relationship between model performance and the amount of training data is possible by analyzing the learning curves. Through cross-validation, it is possible to use k-folds to create a learning curve, train the model on different subsets of data, and evaluate its performance on the validation set. Cross-validation is a technique used to assess the performance of a model by dividing data into k-folds or

**Table 7 Random forest's and gradient boosting's performance in predicting if a document is related to fraud was evaluated using the area under the curve (AUC).** T1, T2, T3, and T4 are the corresponding datasets for the four contexts where a subject obtained from CorEx predominates.

| Classification method's | Predictive accuracy | | | | Mean |
|---|---|---|---|---|---|
| | T1 | T2 | T3 | T4 | |
| Random forest: AUC | 0.91 | 0.90 | 0.89 | 0.77 | 0.87 |
| Gradient boosting: AUC | 0.91 | 0.92 | 0.90 | 0.80 | 0.88 |

k-subsets; This allows the model to be trained and evaluated k times, each time on a different subset of data. On the other hand, ROC (receiver operating characteristics) curves provide a way to assess the trade-off between model sensitivity and specificity, so they can help determine the optimal threshold for classification tasks. Together, these metrics provide a comprehensive approach to assess and validate the performance of a machine learning model.

Using multiple datasets to validate a model contributes to a more robust estimate of its performance. In this context, four datasets WebScraping, Students, NN, and ChatGPT, were used to perform the tests. Through the application of learning curves, the classifiers (RF and GB) were trained with the four datasets individually. For their validation, the three remaining sets were concatenated, all for each of the four study topics. In other words, four training-validation rounds were carried out, one per dataset; for example, for the first set of tests, the model was trained with WebScraping and validated with (Students+NN+ChatGPT), for each topic, for the three rounds. The remaining datasets were exchanged until all possible combinations were covered. As can be seen in Fig. 3, a recurring behavior was identified because of applying this technique, observing in the different test rounds that GB has a low bias and acceptable variance in the four topics, which suggests that the model adequately works both in the training set and the test set. Therefore, it can capture the relationship between the characteristics and the objective variable. That is, it does not make assumptions. Furthermore, the model is not sensitive to variations in the training set and can generalize new data well. In the case of RF, it can be mentioned that, in contrast, it has a high bias and a high variance, which means that the model cannot efficiently capture the relationship between the characteristics and the target variable in the dataset and is sensitive to variations in the dataset so it cannot generalize well to new data.

Each of the four datasets was used to train GB once it was verified that its performance was superior to RF. In contrast, the remaining three sets were used to test the model's performance. This process was repeated using each of the four training datasets and testing the performance with the remaining three until all possible training-test combinations were covered in the four established topics. ROC curves were generated for each training-test combination to assess the model's performance; This made it possible to compare the classifier's performance on different datasets and determine which dataset reported superior performance.

To identify the behavior the different combinations provide, we can look at the AUC scores obtained for each topic using GB in the four assessments. The higher the AUC score,

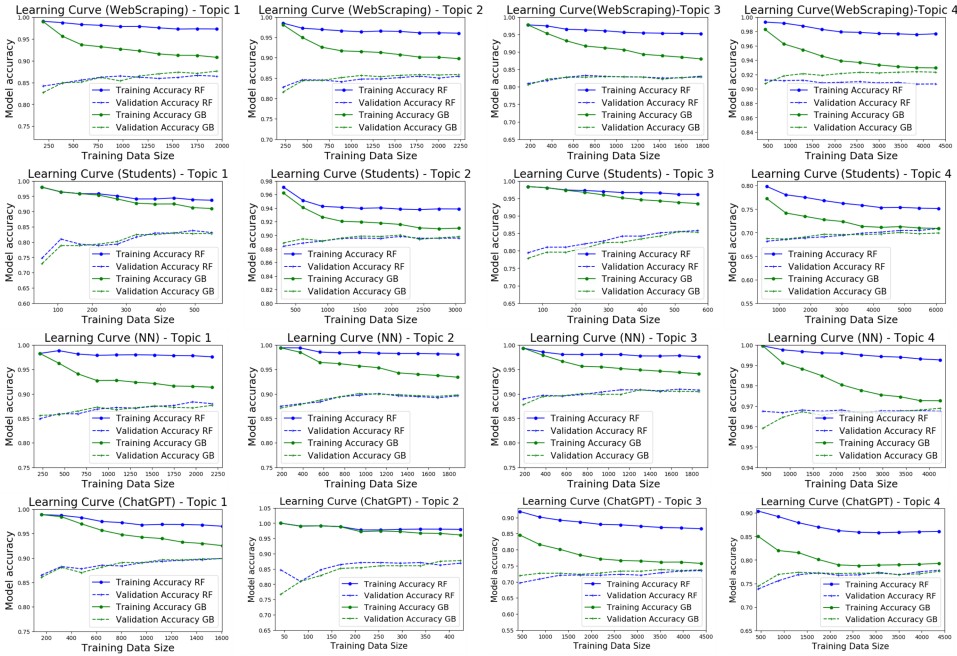

**Figure 3   Learning curves for the four tests were carried out using RF and GB models.** This figure also shows the training time of the different models as a function of the size of the training set.

the better the performance of the classifier. As can be seen in the Tables 8, 9, 10 and 11, the combinations that present the best performance are those where the dataset used for training is the same used for testing in all evaluations, obtaining consistently high AUC scores in all the topics, values represented by the main diagonal of each matrix, highlighted in bold. In those tests where the datasets with which the model was tested differ from those with which the model was trained, we observed that the metrics obtained fluctuate in the combinations made; some have higher AUC values for specific topics and classifiers, while others have lower scores, this suggests that the performance of the classifiers depends on the dataset used for training and testing. Additionally, the values of the four tests were averaged, as seen in Table 12, observing the same behavior. In addition, we use the cross-validation (CV) technique to contrast the data obtained with the external validation. The dataset was divided into five different "folds," allowing us to train and test the model iteratively. Each iteration used a different fold as a test set, while the remaining folds were used for training. Once all the iterations were completed, the results were averaged to derive a comprehensive performance measure for the model, as seen in Table 13. In this context, it is possible to affirm that the model is generalizable since it has been externally validated and cross-validated using the study datasets and the best performance classifier, in addition to the scores in a general way in the different phases of training and test, per topic are consistently high.

**Table 8 Topic 1.**

|  | | | | Test | | |
|---|---|---|---|---|---|---|
|  | Dataset | WS | NN | ST | Chat | Dif |
| Train | WS | 0.92 | 0.92 | 0.77 | 0.87 | 0.15 |
|  | NN | 0.90 | 0.92 | 0.77 | 0.87 | 0.13 |
|  | ST | 0.84 | 0.84 | 0.85 | 0.79 | 0.06 |
|  | Chat | 0.90 | 0.82 | 0.83 | 0.93 | 0.11 |

**Table 9 Topic 2.**

|  | | | | Test | | |
|---|---|---|---|---|---|---|
|  | Dataset | WS | NN | ST | Chat | Dif |
| Train | WS | 0.93 | 0.92 | 0.89 | 0.83 | 0.10 |
|  | NN | 0.90 | 0.89 | 0.86 | 0.80 | 0.10 |
|  | ST | 0.87 | 0.86 | 0.93 | 0.89 | 0.07 |
|  | Chat | 0.81 | 0.84 | 0.84 | 0.83 | 0.03 |

**Table 10 Topic 3.**

|  | | | | Test | | |
|---|---|---|---|---|---|---|
|  | Dataset | WS | NN | ST | Chat | Dif |
| Train | WS | 0.91 | 0.91 | 0.83 | 0.76 | 0.15 |
|  | NN | 0.88 | 0.91 | 0.82 | 0.76 | 0.15 |
|  | ST | 0.82 | 0.77 | 0.89 | 0.80 | 0.12 |
|  | Chat | 0.83 | 0.80 | 0.80 | 0.82 | 0.03 |

## *Discussion*

This section compares different topic modeling approaches to capture fraud-related phrases and their computational complexity. The distribution of the main terms and topics obtained from the classic LDA is presented in Table 3, with words related to fraud labeled in a specific format. However, fraud-related words are randomly distributed in the topics without any specific clustering, which prevents tagging the topics with the vertices of the FTT; this suggests that the modeling approach cannot determine the relationship between fraud and the FTT. As a result, it cannot be applied on this initial attempt.

Like its unsupervised LDA predecessor, guided LDA does not show any visible grouping by topic and cannot be associated with the FTT. However, the CorEx algorithm performs highly satisfactorily with grouping words by themes. Table 4 shows how words are arranged by a specific format related to the vertex of a fraud triangle, allowing for labeling based on their theme of "pressure, opportunity, and rationalization"; this allows a connection between the FTT and the results obtained by the CorEx model, suggesting that fraud-related phrases within the same individual and corresponding to topics related to the fraud triangle indicate potential fraudsters requiring further investigation.

**Table 11  Topic 4.**

|  | | Test | | | | |
|  | Dataset | WS | NN | ST | Chat | Dif |
|---|---|---|---|---|---|---|
| Train | WS | 0.82 | 0.82 | 0.66 | 0.68 | 0.16 |
|  | NN | 0.79 | 0.82 | 0.65 | 0.69 | 0.17 |
|  | ST | 0.60 | 0.60 | 0.78 | 0.70 | 0.18 |
|  | Chat | 0.67 | 0.64 | 0.67 | 0.76 | 0.12 |

**Table 12  Average of the four tests per topic.**

|  | | Test | | | | |
|  | Dataset | WS | NN | ST | Chat | Dif |
|---|---|---|---|---|---|---|
| Train | WS | 0.90 | 0.89 | 0.79 | 0.78 | 0.12 |
|  | NN | 0.87 | 0.89 | 0.78 | 0.78 | 0.11 |
|  | ST | 0.78 | 0.77 | 0.86 | 0.80 | 0.10 |
|  | Chat | 0.80 | 0.78 | 0.79 | 0.84 | 0.06 |

**Table 13  Average cross validation (CV) scores.**

| Topics | Folds | | | | | CV Score |
|  | 1 | 2 | 3 | 4 | 5 | |
|---|---|---|---|---|---|---|
| 1: | 0.85 | 0.85 | 0.84 | 0.84 | 0.86 | 0.85 |
| 2: | 0.86 | 0.84 | 0.85 | 0.85 | 0.85 | 0.85 |
| 3: | 0.79 | 0.78 | 0.77 | 0.78 | 0.77 | 0.78 |
| 4: | 0.80 | 0.79 | 0.79 | 0.79 | 0.80 | 0.79 |

The dataset is balanced between fraud and non-fraud classes. It is mentioned that analyzing the results with balanced precision or the area under the curve (AUC-ROC) is preferable when dealing with imbalanced data. CorEx shows higher recall, meaning it finds more true positives but has a lower precision or higher false positive rate. Additionally, CorEx outperforms normal and GuidedLDA in terms of balanced accuracy. The semi-supervised approach is considered an alternative strategy to the classic unsupervised model, as it avoids challenges in determining the nature of topics and their labels. Although the topics identified by CorEx do not cover all fraud theories, they align with factors in the FTT, such as "pressure, opportunity, and rationalization." This approach, incorporating semi-supervised topic modeling techniques and pre-obtained keywords from LDA, is beneficial for identifying relevant topics.

To analyze the computational complexity of LDA, GuidedLDA, and CorEx approaches in discovering latent themes and structures in data, it is essential to understand the practical implications and considerations that researchers should consider when choosing one of these approaches. In the case of LDA, its complexity is influenced by critical factors such as the number of documents (N), the size of the vocabulary (V), and the number of topics (K), with an approximate complexity of O ($I * N * K * V$) (*Ihou & Bouguila, 2019*), where I

represents the number of iterations that an algorithm requires to converge or reach a steady state. The Expectation-Maximization (EM) algorithm used in LDA refers to the number of times the expectation and maximization steps are performed to fit the model to the data. The complexity grows as the number of documents, the size of the vocabulary, and the number of topics increase. This behavior can limit the scalability of LDA on massive data sets or in situations where a high level of thematic granularity is sought. On the other hand, GuidedLDA, by incorporating external information to guide topic assignments, can present additional complexity due to the extra computations required to integrate these guides. However, it follows a similar structure to the LDA in terms of complexity. The benefits of the guide can be remarkable, especially in cases where the interpretability and quality of the topics are a priority. However, an increase in training time can accompany this improvement. In contrast, CorEx differs from other techniques by addressing the correlation and dependency between variables, which impacts its complexity depending on the dimensionality and the number of samples. Since CorEx operates differently from the probabilistic approach of LDA and GuidedLDA, its complexity is influenced by the number of samples without a fixed number of topic parameters. In summary, the choice between these approaches must consider not only the quality of the results but also the computational complexity and characteristics of the data in question.

## CONCLUSIONS

Fraud study and investigation are critical in addressing social disorder and the security threat it poses to government and business. To effectively combat fraud, it is essential to deepen the analysis of fraudulent activities and develop proactive identification strategies. This research used topic modeling and Machine learning techniques, focusing on the FTT and using various study corpora. The generation of four datasets was necessary due to the scarcity of resources in this field and the need for fraud-related information. Applying a semi-supervised approach to theme modeling, using the CorEx and GuidedLDA algorithms, demonstrated that CorEx was more successful in creating consistent and interpretable themes aligned with the vertices of the fraud triangle. The probabilities of the document-subject associations extracted from the models were then used as input for the gradient boosting and random forest classification methods to predict fraud-related behaviors. Evaluation of the model's performance using ROC curves and the AUC metric revealed that gradient boosting slightly outperformed random forest, achieving an average classification accuracy of 88% compared to 87%; This represents a 7% improvement over the results obtained in a previous study (*Sánchez-Aguayo, Urquiza-Aguiar & Estrada-Jiménez, 2022*). Semi-supervised approaches like CorEx in text mining contribute to a better analysis of the combination of expertise and domain scalability. Using multiple datasets to test the model's performance yielded promising results, indicating that the model can be generalized. In addition, the model obtained a low bias and an acceptable variance in the four subjects, which indicates good performance in the training and test sets.

## Future work

In future work, it is proposed to apply new approaches concerning topic modeling, such as BerTopic, to improve the identification and analysis of relevant topics in large data sets. In this sense, data sets with more information should be generated. This new approach could involve advanced deep learning techniques, such as convolutional neural networks or recurrent neural networks, allowing a more precise and contextualized representation of data documents. In addition, incorporating multimodal information, such as images or videos, into topic modeling could be investigated, enriching the understanding of topics by considering different data modalities. In summary, further study and analysis of topic modeling promise innovative approaches that will improve the ability to identify and analyze topics in large volumes of data more accurately.

### Funding

Marco Sánchez is a recipient of a teaching assistant fellowship from Escuela Politecnica Nacional for doctoral studies in computer science. The funders had no role in study design, data collection and analysis, decision to publish, or preparation of the manuscript.

### Grant Disclosures

The following grant information was disclosed by the authors:
Escuela Politecnica Nacional for doctoral studies in computer science.

### Competing Interests

The authors declare there are no competing interests.

### Author Contributions

- Marco Sánchez conceived and designed the experiments, performed the experiments, analyzed the data, performed the computation work, prepared figures and/or tables, authored or reviewed drafts of the article, and approved the final draft.
- Luis Urquiza conceived and designed the experiments, analyzed the data, authored or reviewed drafts of the article, and approved the final draft.

### Data Availability

  The raw data is available in the Supplemental File.

### Supplemental Information

Supplemental information for this article can be found online at http://dx.doi.org/10.7717/peerj-cs.1733#supplemental-information.

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
