# Peer review of "Improving fraud detection with semi-supervised topic modeling and keyword integration"

_PeerJ Computer Science, doi:10.7717/peerj-cs.1733_

## Round 0.1 · original submission · Major Revisions

I have received the review reports for your paper submitted to PeerJ Computer Science from the reviewers. According to the reports, I will recommend major revision to your paper. Please refer to the reviewers’ opinions to improve your paper. Please also write a revision note such that the reviewers can easily check whether their comments are fully addressed. We look forward to receiving your revised manuscript soon.
With best regards

Reviewer 1 ·

Basic reporting

See comments in pdf file. The article would benefit from a strong proof reading. Article can be shortened by removing a section that is not needed. Synthetic data is not shared.

Experimental design

No major concerns, see comments in the document for one additional experiment which would help determine if the approach is generalizable.

Validity of the findings

Mostly ok. Data and code are missing in a few cases (some data might be in the Jupyter notebook but hard to extract from there). One overstatement of the results which should be fixed.

Additional comments

Generally very interesting work on topic modeling and addresses the many concerns that others have stated about interpreting the topics generated. This approach is replicable by others in different domains.

Annotated reviews are not available for download in order to protect the identity of reviewers who chose to remain anonymous.

Reviewer 2 ·

Basic reporting

1. In Line 52, authors should provide the reference for ‘fraud triangle theory’.

2. Several places in the manuscript, English quality and grammar need to be enhanced for improvement in readability. For example, in Line 90, the sentence - “Their authors use LDA to find groups of words related to the attributes of the metadata of the study documents” needs to be rewritten.

3. Authors must ensure that the acronyms used throughout the manuscript are defined prior to their usage.

4. In line 137-139, section numbers are missing.

5. Typo error in Line 212, ‘GuiedLDA’ should be ‘GuidedLDA’.

6. Authors should write the ‘conclusion’ section in more of a conclusive tone.

Experimental design

1. In Related Work section, provide a table consisting of method, year of publication, computational benefits and limitations for significant state-of-the-art.

2. Authors should provide a paragraph for explanation of Figure 1. The flow among the computational modules needs to be described.

Validity of the findings

1. Authors should include a comprehensive discussion about the computational and amortized complexity analysis for the proposed framework. This discussion shall serve as a fundamental component of this work that involves algorithm design, optimization, or performance evaluation. The analysis provides valuable insights into the efficiency and scalability of the proposed framework, shedding light on its practicality and potential real-world applications.

Additional comments

1. Authors must ensure that each reference in the list has vol, issue, number, pages info.

Reviewer 3 ·

Basic reporting

1. The manuscript is written in adequate English without significant typos/errors. Line 137-140 missed section numbers.
2. The literature review is sufficient and relevant to this work.
3. The structure is well-organized and meets the standard of scientific articles.
4. This work presents interesting results which outperform comparing methods.

Experimental design

1. Several synthetic datasets were used. The datasets are popular and representative.
2. The adopted methods for topic modeling and classification are common. A novel customized topic modeling method CorEx was used to identify the probabilities that the corpus documents belong to a topic.

Validity of the findings

1. The work shows how the fraud triangle theory can be integrated into CorEx through ”keywords” related to the vertices of this theory. They also show that CorEx produces more relevant topics than its unsupervised and semi-supervised variants of LDA.
2. The model was tested in multiple conditions to ensure it worked reliably in all situations, confirming that it could accurately predict outcomes in various contexts. The results of these tests were then used to validate the model’s performance and provide evidence of its accuracy.
3. Evaluating the model’s performance using ROC curves and its AUC metric revealed that the Gradient Boosting is slightly more efficient in classifying fraud cases with an average of 88% vs. 87% of Random Forest. The results demonstrate an improvement of 7% over previous work.

Additional comments

Overall speaking, this work is well-written with proven results. I will suggest acceptance of this work.

---

## Round 0.2 · accepted · Accept

We are happy to inform you that your manuscript has been accepted for publication since the reviewers' comments had been answered.

Reviewer 2 ·

Basic reporting

Authors have incorporated the revisions in the updated version of the manuscript. No further revisions are suggested now.

Experimental design

Revisions implemented.

Validity of the findings

Revised by the Authors.

Additional comments

Nil